# A Regularized Framework for
# Sparse and Structured Neural Attention

**Vlad Niculae**[*]
Cornell University
Ithaca, NY
vlad@cs.cornell.edu

**Mathieu Blondel**
NTT Communication Science Laboratories
Kyoto, Japan
mathieu@mblondel.org

## Abstract

Modern neural networks are often augmented with an attention mechanism, which tells the network where to focus within the input. We propose in this paper a new framework for sparse and structured attention, building upon a smoothed max operator. We show that the gradient of this operator defines a mapping from real values to probabilities, suitable as an attention mechanism. Our framework includes softmax and a slight generalization of the recently-proposed sparsemax as special cases. However, we also show how our framework can incorporate modern structured penalties, resulting in more interpretable attention mechanisms, that focus on entire segments or groups of an input. We derive efficient algorithms to compute the forward and backward passes of our attention mechanisms, enabling their use in a neural network trained with backpropagation. To showcase their potential as a drop-in replacement for existing ones, we evaluate our attention mechanisms on three large-scale tasks: textual entailment, machine translation, and sentence summarization. Our attention mechanisms improve interpretability without sacrificing performance; notably, on textual entailment and summarization, we outperform the standard attention mechanisms based on softmax and sparsemax.

## 1 Introduction

Modern neural network architectures are commonly augmented with an attention mechanism, which tells the network where to look within the input in order to make the next prediction. Attention-augmented architectures have been successfully applied to machine translation [2, 29], speech recognition [10], image caption generation [44], textual entailment [38, 31], and sentence summarization [39], to name but a few examples. At the heart of attention mechanisms is a mapping function that converts real values to probabilities, encoding the relative importance of elements in the input. For the case of sequence-to-sequence prediction, at each time step of generating the output sequence, attention probabilities are produced, conditioned on the current state of a decoder network. They are then used to aggregate an input representation (a variable-length list of vectors) into a single vector, which is relevant for the current time step. That vector is finally fed into the decoder network to produce the next element in the output sequence. This process is repeated until the end-of-sequence symbol is generated. Importantly, such architectures can be trained end-to-end using backpropagation.

Alongside empirical successes, neural attention—while not necessarily correlated with human attention—is increasingly crucial in bringing more **interpretability** to neural networks by helping explain how individual input elements contribute to the model's decisions. However, the most commonly used attention mechanism, *softmax*, yields dense attention weights: all elements in the input always make at least a small contribution to the decision. To overcome this limitation, *sparsemax* was recently proposed [31], using the Euclidean projection onto the simplex as a sparse alternative to

---

[*]Work performed during an internship at NTT Commmunication Science Laboratories, Kyoto, Japan.

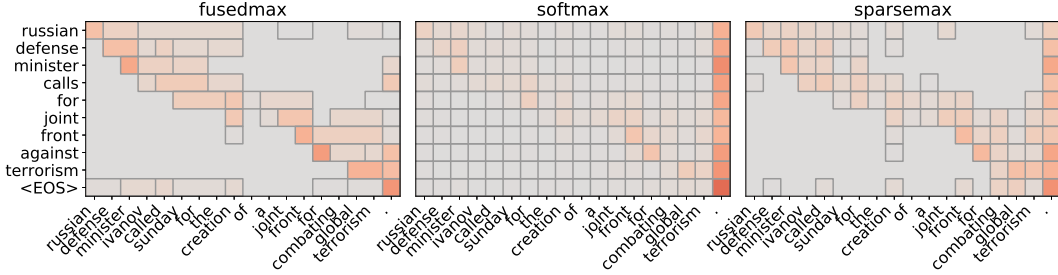

Figure 1: Attention weights produced by the proposed *fusedmax*, compared to *softmax* and *sparsemax*, on sentence summarization. The input sentence to be summarized (taken from [39]) is along the $x$-axis. From top to bottom, each row shows where the attention is distributed when producing each word in the summary. All rows sum to 1, the grey background corresponds to exactly 0 (never achieved by softmax), and adjacent positions with exactly equal weight are not separated by borders. Fusedmax pays attention to contiguous segments of text with equal weight; such segments never occur with softmax and sparsemax. In addition to enhancing interpretability, we show in §4.3 that fusedmax outperforms both softmax and sparsemax on this task in terms of ROUGE scores.

softmax. Compared to softmax, sparsemax outputs more interpretable attention weights, as illustrated in [31] on the task of textual entailment. The principle of parsimony, which states that simple explanations should be preferred over complex ones, is not, however, limited to sparsity: it remains open whether new attention mechanisms can be designed to benefit from more structural prior knowledge.

**Our contributions.** The success of sparsemax motivates us to explore new attention mechanisms that can both output sparse weights and take advantage of structural properties of the input through the use of modern sparsity-inducing penalties. To do so, we make the following contributions:

1) We propose a new **general framework** that builds upon a max operator, regularized with a strongly convex function. We show that this operator is differentiable, and that its gradient defines a mapping from real values to probabilities, suitable as an attention mechanism. Our framework **includes as special cases** both **softmax** and a slight generalization of **sparsemax**. (§2)

2) We show how to incorporate the fused lasso [42] in this framework, to derive a new attention mechanism, named *fusedmax*, which encourages the network to pay attention to **contiguous segments of text** when making a decision. This idea is illustrated in Figure 1 on sentence summarization. For cases when the contiguity assumption is too strict, we show how to incorporate an OSCAR penalty [7] to derive a new attention mechanism, named *oscarmax*, that encourages the network to pay equal attention to **possibly non-contiguous groups of words**. (§3)

3) In order to use attention mechanisms defined under our framework in an autodiff toolkit, two problems must be addressed: evaluating the attention itself and computing its Jacobian. However, our attention mechanisms require solving a convex optimization problem and do not generally enjoy a simple analytical expression, unlike softmax. Computing the Jacobian of the solution of an optimization problem is called argmin/argmax differentiation and is currently an area of active research (cf. [1] and references therein). One of **our key algorithmic contributions is to show how to compute this Jacobian** under our general framework, as well as for fused lasso and OSCAR. (§3)

4) To showcase the potential of our new attention mechanisms as a **drop-in replacement** for existing ones, we show empirically that our new attention mechanisms enhance interpretability while achieving comparable or better accuracy on three diverse and challenging tasks: **textual entailment, machine translation, and sentence summarization**. (§4)

**Notation.** We denote the set $\{1, \ldots, d\}$ by $[d]$. We denote the $(d-1)$-dimensional probability simplex by $\Delta^d := \{\boldsymbol{x} \in \mathbb{R}^d : \|\boldsymbol{x}\|_1 = 1, \boldsymbol{x} \geq 0\}$ and the Euclidean projection onto it by $P_{\Delta^d}(\boldsymbol{x}) := \arg\min_{\boldsymbol{y} \in \Delta^d} \|\boldsymbol{y} - \boldsymbol{x}\|^2$. Given a function $f: \mathbb{R}^d \to \mathbb{R} \cup \{\infty\}$, its convex conjugate is defined by $f^*(\boldsymbol{x}) := \sup_{\boldsymbol{y} \in \text{dom } f} \boldsymbol{y}^{\text{T}} \boldsymbol{x} - f(\boldsymbol{y})$. Given a norm $\|\cdot\|$, its dual is defined by $\|\boldsymbol{x}\|_* := \sup_{\|\boldsymbol{y}\| \leq 1} \boldsymbol{y}^{\text{T}} \boldsymbol{x}$. We denote the subdifferential of a function $f$ at $\boldsymbol{y}$ by $\partial f(\boldsymbol{y})$. Elements of the subdifferential are called subgradients and when $f$ is differentiable, $\partial f(\boldsymbol{y})$ contains a single element, the gradient of $f$ at $\boldsymbol{y}$, denoted by $\nabla f(\boldsymbol{y})$. We denote the Jacobian of a function $g: \mathbb{R}^d \to \mathbb{R}^d$ at $\boldsymbol{y}$ by $J_g(\boldsymbol{y}) \in \mathbb{R}^{d \times d}$ and the Hessian of a function $f: \mathbb{R}^d \to \mathbb{R}$ at $\boldsymbol{y}$ by $H_f(\boldsymbol{y}) \in \mathbb{R}^{d \times d}$.

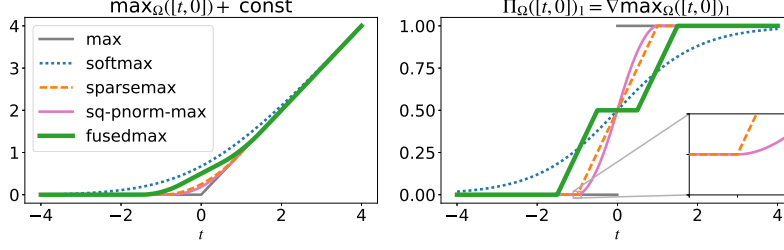

Figure 2: The proposed $\max_\Omega(\boldsymbol{x})$ operator up to a constant (left) and the proposed $\Pi_\Omega(\boldsymbol{x})$ mapping (right), illustrated with $\boldsymbol{x} = [t, 0]$ and $\gamma = 1$. In this case, $\max_\Omega(\boldsymbol{x})$ is a ReLu-like function and $\Pi_\Omega(\boldsymbol{x})$ is a sigmoid-like function. Our framework recovers *softmax* (negative entropy) and *sparsemax* (squared 2-norm) as special cases. We also introduce three new attention mechanisms: *sq-pnorm-max* (squared $p$-norm, here illustrated with $p = 1.5$), *fusedmax* (squared 2-norm + fused lasso), and *oscarmax* (squared 2-norm + OSCAR; not pictured since it is equivalent to fusedmax in 2-d). Except for softmax, which never exactly reaches 0, all mappings shown on the right encourage sparse outputs.

## 2 Proposed regularized attention framework

### 2.1 The max operator and its subgradient mapping

To motivate our proposal, we first show in this section that the subgradients of the maximum operator define a mapping from $\mathbb{R}^d$ to $\Delta^d$, but that this mapping is highly unsuitable as an attention mechanism. The maximum operator is a function from $\mathbb{R}^d$ to $\mathbb{R}$ and can be defined by

$$\max(\boldsymbol{x}) \coloneqq \max_{i \in [d]} x_i = \sup_{\boldsymbol{y} \in \Delta^d} \boldsymbol{y}^{\mathrm{T}} \boldsymbol{x}.$$

The equality on the r.h.s comes from the fact that the supremum of a linear form over the simplex is always achieved at one of the vertices, i.e., one of the standard basis vectors $\{\boldsymbol{e}_i\}_{i=1}^d$. Moreover, it is not hard to check that any solution $\boldsymbol{y}^\star$ of that supremum is precisely a subgradient of $\max(\boldsymbol{x})$: $\partial \max(\boldsymbol{x}) = \{\boldsymbol{e}_{i^\star} : i^\star \in \arg\max_{i \in [d]} x_i\}$. We can see these subgradients as a mapping $\Pi \colon \mathbb{R}^d \to \Delta^d$ that **puts all the probability mass onto a single element**: $\Pi(\boldsymbol{x}) = \boldsymbol{e}_i$ for any $\boldsymbol{e}_i \in \partial \max(\boldsymbol{x})$. However, this behavior is undesirable, as the resulting mapping is a discontinuous function (a Heaviside step function when $\boldsymbol{x} = [t, 0]$), which is not amenable to optimization by gradient descent.

### 2.2 A regularized max operator and its gradient mapping

These shortcomings encourage us to consider a regularization of the maximum operator. Inspired by the seminal work of Nesterov [35], we apply a smoothing technique. The conjugate of $\max(\boldsymbol{x})$ is

$$\max{}^*(\boldsymbol{y}) = \begin{cases} 0, & \text{if } \boldsymbol{y} \in \Delta^d \\ \infty, & \text{o.w.} \end{cases}.$$

For a proof, see for instance [33, Appendix B]. We now add regularization to the conjugate

$$\max{}^*_\Omega(\boldsymbol{y}) \coloneqq \begin{cases} \gamma\Omega(\boldsymbol{y}), & \text{if } \boldsymbol{y} \in \Delta^d \\ \infty, & \text{o.w.} \end{cases},$$

where we assume that $\Omega \colon \mathbb{R}^d \to \mathbb{R}$ is $\beta$-**strongly convex** w.r.t. some norm $\|\cdot\|$ and $\gamma > 0$ controls the regularization strength. To define a smoothed $\max$ operator, we take the conjugate once again

$$\max{}_\Omega(\boldsymbol{x}) = \max{}^{**}_\Omega(\boldsymbol{x}) = \sup_{\boldsymbol{y} \in \mathbb{R}^d} \boldsymbol{y}^{\mathrm{T}} \boldsymbol{x} - \max{}^*_\Omega(\boldsymbol{y}) = \sup_{\boldsymbol{y} \in \Delta^d} \boldsymbol{y}^{\mathrm{T}} \boldsymbol{x} - \gamma\Omega(\boldsymbol{y}). \tag{1}$$

Our main proposal is a mapping $\Pi_\Omega \colon \mathbb{R}^d \to \Delta^d$, defined as the *argument* that achieves this supremum.

$$\boxed{\Pi_\Omega(\boldsymbol{x}) \coloneqq \arg\max_{\boldsymbol{y} \in \Delta^d} \boldsymbol{y}^{\mathrm{T}} \boldsymbol{x} - \gamma\Omega(\boldsymbol{y}) = \nabla\max{}_\Omega(\boldsymbol{x})}$$

The r.h.s. holds by combining that i) $\max_\Omega(\boldsymbol{x}) = (\boldsymbol{y}^\star)^{\mathrm{T}}\boldsymbol{x} - \max{}^*_\Omega(\boldsymbol{y}^\star) \Leftrightarrow \boldsymbol{y}^\star \in \partial\max_\Omega(\boldsymbol{x})$ and ii) $\partial\max_\Omega(\boldsymbol{x}) = \{\nabla\max_\Omega(\boldsymbol{x})\}$, since (1) has a unique solution. Therefore, $\Pi_\Omega$ is a **gradient mapping**. We illustrate $\max_\Omega$ and $\Pi_\Omega$ for various choices of $\Omega$ in Figure 2 (2-d) and in Appendix C.1 (3-d).

**Importance of strong convexity.** Our $\beta$-strong convexity assumption on $\Omega$ plays a crucial role and should not be underestimated. Recall that a function $f\colon \mathbb{R}^d \to \mathbb{R}$ is $\beta$-strongly convex w.r.t. a norm $\|\cdot\|$ if and only if its conjugate $f^*$ is $\frac{1}{\beta}$-smooth w.r.t. the dual norm $\|\cdot\|_*$ [46, Corollary 3.5.11] [22, Theorem 3]. This is sufficient to ensure that $\max_\Omega$ is $\frac{1}{\gamma\beta}$**-smooth**, or, in other words, that it is differentiable everywhere and its gradient, $\Pi_\Omega$, is $\frac{1}{\gamma\beta}$-Lipschitz continuous w.r.t. $\|\cdot\|_*$.

**Training by backpropagation.** In order to use $\Pi_\Omega$ in a neural network trained by backpropagation, two problems must be addressed for any regularizer $\Omega$. The first is the **forward** computation: how to evaluate $\Pi_\Omega(\boldsymbol{x})$, i.e., how to solve the optimization problem in (1). The second is the **backward** computation: how to evaluate the Jacobian of $\Pi_\Omega(\boldsymbol{x})$, or, equivalently, the Hessian of $\max_\Omega(\boldsymbol{x})$. One of our key contributions, presented in §3, is to show how to solve these two problems for general differentiable $\Omega$, as well as for two structured regularizers: fused lasso and OSCAR.

## 2.3 Recovering softmax and sparsemax as special cases

Before deriving new attention mechanisms using our framework, we now show how we can recover softmax and sparsemax, using a specific regularizer $\Omega$.

**Softmax.** We choose $\Omega(\boldsymbol{y}) = \sum_{i=1}^d y_i \log y_i$, the negative entropy. The conjugate of the negative entropy restricted to the simplex is the $\log\text{sum}\exp$ [9, Example 3.25]. Moreover, if $f(\boldsymbol{x}) = \gamma g(\boldsymbol{x})$ for $\gamma > 0$, then $f^*(\boldsymbol{y}) = \gamma g^*(\boldsymbol{y}/\gamma)$. We therefore get a closed-form expression: $\max_\Omega(\boldsymbol{x}) = \gamma \ \log\text{sum}\exp(\boldsymbol{x}/\gamma) \coloneqq \gamma \log \sum_{i=1}^d e^{x_i/\gamma}$. Since the negative entropy is 1-strongly convex w.r.t. $\|\cdot\|_1$ over $\Delta^d$, we get that $\max_\Omega$ is $\frac{1}{\gamma}$-smooth w.r.t. $\|\cdot\|_\infty$. We obtain the classical softmax, with temperature parameter $\gamma$, by taking the gradient of $\max_\Omega(\boldsymbol{x})$,

$$\Pi_\Omega(\boldsymbol{x}) = \frac{e^{\boldsymbol{x}/\gamma}}{\sum_{i=1}^d e^{x_i/\gamma}}, \qquad \text{(softmax)}$$

where $e^{\boldsymbol{x}/\gamma}$ is evaluated element-wise. Note that some authors also call $\max_\Omega$ a "soft max." Although $\Pi_\Omega$ is really a soft *arg max*, we opt to follow the more popular terminology. When $\boldsymbol{x} = [t, 0]$, it can be checked that $\max_\Omega(\boldsymbol{x})$ reduces to the softplus [16] and $\Pi_\Omega(\boldsymbol{x})_1$ to a sigmoid.

**Sparsemax.** We choose $\Omega(\boldsymbol{y}) = \frac{1}{2}\|\boldsymbol{y}\|_2^2$, also known as Moreau-Yosida regularization in proximal operator theory [35, 36]. Since $\frac{1}{2}\|\boldsymbol{y}\|_2^2$ is 1-strongly convex w.r.t. $\|\cdot\|_2$, we get that $\max_\Omega$ is $\frac{1}{\gamma}$-smooth w.r.t. $\|\cdot\|_2$. In addition, it is easy to verify that

$$\Pi_\Omega(\boldsymbol{x}) = P_{\Delta^d}(\boldsymbol{x}/\gamma) = \operatorname*{arg\,min}_{\boldsymbol{y} \in \Delta^d} \|\boldsymbol{y} - \boldsymbol{x}/\gamma\|^2. \qquad \text{(sparsemax)}$$

This mapping was introduced *as is* in [31] with $\gamma = 1$ and was named sparsemax, due to the fact that it is a sparse alternative to softmax. Our derivation thus gives us a slight generalization, where $\gamma$ controls the sparsity (the smaller, the sparser) and could be tuned; in our experiments, however, we follow the literature and set $\gamma = 1$. The Euclidean projection onto the simplex, $P_{\Delta^d}$, can be computed exactly [34, 15] (we discuss the complexity in Appendix B). Following [31], the Jacobian of $\Pi_\Omega$ is

$$J_{\Pi_\Omega}(\boldsymbol{x}) = \frac{1}{\gamma} J_{P_{\Delta^d}}(\boldsymbol{x}/\gamma) = \frac{1}{\gamma} \left( \operatorname{diag}(\boldsymbol{s}) - \boldsymbol{s}\boldsymbol{s}^{\mathrm{T}}/\|\boldsymbol{s}\|_1 \right),$$

where $\boldsymbol{s} \in \{0,1\}^d$ indicates the nonzero elements of $\Pi_\Omega(\boldsymbol{x})$. Since $\Pi_\Omega$ is Lipschitz continuous, Rademacher's theorem implies that $\Pi_\Omega$ is differentiable almost everywhere. For points where $\Pi_\Omega$ is not differentiable (where $\max_\Omega$ is not twice differentiable), we can take an arbitrary matrix in the set of Clarke's generalized Jacobians [11], the convex hull of Jacobians of the form $\lim_{\boldsymbol{x}_t \to \boldsymbol{x}} J_{\Pi_\Omega}(\boldsymbol{x}_t)$ [31].

## 3 Deriving new sparse and structured attention mechanisms

### 3.1 Differentiable regularizer $\Omega$

Before tackling more structured regularizers, we address in this section the case of general differentiable regularizer $\Omega$. Because $\Pi_\Omega(\boldsymbol{x})$ involves maximizing (1), a concave function over the simplex, it can be computed globally using any off-the-shelf projected gradient solver. Therefore, the main challenge is how to compute the Jacobian of $\Pi_\Omega$. This is what we address in the next proposition.

**Proposition 1** *Jacobian of $\Pi_\Omega$ for any differentiable $\Omega$ (backward computation)*

*Assume that $\Omega$ is differentiable over $\Delta^d$ and that $\Pi_\Omega(\boldsymbol{x}) = \arg\max_{\boldsymbol{y}\in\Delta^d} \boldsymbol{y}^{\mathrm{T}}\boldsymbol{x} - \gamma\Omega(\boldsymbol{y}) = \boldsymbol{y}^\star$ has been computed. Then the Jacobian of $\Pi_\Omega$ at $\boldsymbol{x}$, denoted $J_{\Pi_\Omega}$, can be obtained by solving the system*
$$(I + A(B - I))\, J_{\Pi_\Omega} = A,$$
*where we defined the shorthands $A := J_{P_{\Delta^d}}(\boldsymbol{y}^\star - \gamma\nabla\Omega(\boldsymbol{y}^\star) + \boldsymbol{x})$ and $B := \gamma H_\Omega(\boldsymbol{y}^\star)$.*

The proof is given in Appendix A.1. Unlike recent work tackling argmin differentiation through matrix differential calculus on the Karush–Kuhn–Tucker (KKT) conditions [1], our proof technique relies on differentiating the fixed point iteration $\boldsymbol{y}^* = P_{\Delta^d}(\boldsymbol{y}^\star - \nabla f(\boldsymbol{y}^\star))$. To compute $J_{\Pi_\Omega}\boldsymbol{v}$ for an arbitrary $\boldsymbol{v} \in \mathbb{R}^d$, as required by backpropagation, we may directly solve $(I + A(B - I))\,(J_{\Pi_\Omega}\boldsymbol{v}) = A\boldsymbol{v}$. We show in Appendix B how this system can be solved efficiently thanks to the structure of $A$.

**Squared $p$-norms.** As a useful example of a differentiable function over the simplex, we consider squared $p$-norms: $\Omega(\boldsymbol{y}) = \frac{1}{2}\|\boldsymbol{y}\|_p^2 = \left(\sum_{i=1}^d y_i^p\right)^{2/p}$, where $\boldsymbol{y}\in\Delta^d$ and $p\in(1,2]$. For this choice of $p$, it is known that the squared p-norm is strongly convex w.r.t. $\|\cdot\|_p$ [3]. This implies that $\max_\Omega$ is $\frac{1}{\gamma(p-1)}$ smooth w.r.t. $\|.\|_q$, where $\frac{1}{p} + \frac{1}{q} = 1$. We call the induced mapping function *sq-pnorm-max*:
$$\Pi_\Omega(\boldsymbol{x}) = \arg\min_{\boldsymbol{y}\in\Delta^d} \frac{\gamma}{2}\|\boldsymbol{y}\|_p^2 - \boldsymbol{y}^{\mathrm{T}}\boldsymbol{x}. \qquad \text{(sq-pnorm-max)}$$
The gradient and Hessian needed for Proposition 1 can be computed by $\nabla\Omega(\boldsymbol{y}) = \frac{\boldsymbol{y}^{p-1}}{\|\boldsymbol{y}\|_p^{p-2}}$ and
$$H_\Omega(\boldsymbol{y}) = \mathrm{diag}(\boldsymbol{d}) + \boldsymbol{u}\boldsymbol{u}^{\mathrm{T}}, \quad \text{where} \quad \boldsymbol{d} = \frac{(p-1)}{\|\boldsymbol{y}\|_p^{p-2}}\,\boldsymbol{y}^{p-2} \quad \text{and} \quad \boldsymbol{u} = \sqrt{\frac{(2-p)}{\|\boldsymbol{y}\|_p^{2p-2}}}\,\boldsymbol{y}^{p-1},$$
with the exponentiation performed element-wise. sq-pnorm-max recovers sparsemax with $p = 2$ and, like sparsemax, encourages sparse outputs. However, as can be seen in the zoomed box in Figure 2 (right), the transition between $\boldsymbol{y}^\star = [0, 1]$ and $\boldsymbol{y}^\star = [1, 0]$ can be smoother when $1 < p < 2$. Throughout our experiments, we use $p = 1.5$.

## 3.2 Structured regularizers: fused lasso and OSCAR

**Fusedmax.** For cases when the input is sequential and the order is meaningful, as is the case for many natural languages, we propose *fusedmax*, an attention mechanism based on *fused lasso* [42], also known as 1-d total variation (TV). Fusedmax encourages paying attention to **contiguous segments**, with equal weights within each one. It is expressed under our framework by choosing $\Omega(\boldsymbol{y}) = \frac{1}{2}\|\boldsymbol{y}\|_2^2 + \lambda\sum_{i=1}^{d-1}|y_{i+1} - y_i|$, i.e., the sum of a strongly convex term and of a 1-d TV penalty. It is easy to verify that this choice yields the mapping
$$\Pi_\Omega(\boldsymbol{x}) = \arg\min_{\boldsymbol{y}\in\Delta^d} \frac{1}{2}\|\boldsymbol{y} - \boldsymbol{x}/\gamma\|^2 + \lambda\sum_{i=1}^{d-1}|y_{i+1} - y_i|. \qquad \text{(fusedmax)}$$

**Oscarmax.** For situations where the contiguity assumption may be too strict, we propose *oscarmax*, based on the OSCAR penalty [7], to encourage attention weights to **merge into clusters with the same value**, regardless of position in the sequence. This is accomplished by replacing the 1-d TV penalty in fusedmax with an $\infty$-norm penalty on each pair of attention weights, i.e., $\Omega(\boldsymbol{y}) = \frac{1}{2}\|\boldsymbol{y}\|_2^2 + \lambda\sum_{i<j}\max(|y_i|, |y_j|)$. This results in the mapping
$$\Pi_\Omega(\boldsymbol{x}) = \arg\min_{\boldsymbol{y}\in\Delta^d} \frac{1}{2}\|\boldsymbol{y} - \boldsymbol{x}/\gamma\|^2 + \lambda\sum_{i<j}\max(|y_i|, |y_j|). \qquad \text{(oscarmax)}$$

**Forward computation.** Due to the $\boldsymbol{y}\in\Delta^d$ constraint, computing fusedmax/oscarmax does not seem trivial on first sight. The next proposition shows how to do so, without any iterative method.

**Proposition 2** *Computing fusedmax and oscarmax (forward computation)*

*fusedmax:* $\Pi_\Omega(\boldsymbol{x}) = P_{\Delta^d}\left(P_{TV}\left(\boldsymbol{x}/\gamma\right)\right), \quad P_{TV}(\boldsymbol{x}) := \arg\min_{\boldsymbol{y}\in\mathbb{R}^d} \frac{1}{2}\|\boldsymbol{y} - \boldsymbol{x}\|^2 + \lambda\sum_{i=1}^{d-1}|y_{i+1} - y_i|.$

*oscarmax:* $\Pi_\Omega(\boldsymbol{x}) = P_{\Delta^d}\left(P_{OSC}\left(\boldsymbol{x}/\gamma\right)\right), P_{OSC}(\boldsymbol{x}) := \arg\min_{\boldsymbol{y}\in\mathbb{R}^d} \frac{1}{2}\|\boldsymbol{y} - \boldsymbol{x}\|^2 + \lambda\sum_{i<j}\max(|y_i|, |y_j|).$

Here, $P_{\text{TV}}$ and $P_{\text{OSC}}$ indicate the proximal operators of 1-d TV and OSCAR, and can be computed **exactly** by [13] and [47], respectively. To remind the reader, $P_{\Delta^d}$ denotes the Euclidean projection onto the simplex and can be computed exactly using [34, 15]. Proposition 2 shows that we can compute fusedmax and oscarmax using the composition of two functions, for which exact non-iterative algorithms exist. This is a surprising result, since the proximal operator of the sum of two functions is not, in general, the composition of the proximal operators of each function. The proof follows by showing that the indicator function of $\Delta^d$ satisfies the conditions of [45, Corollaries 4,5].

**Groups induced by $P_{\text{TV}}$ and $P_{\text{OSC}}$.** Let $\boldsymbol{z}^\star$ be the optimal solution of $P_{\text{TV}}(\boldsymbol{x})$ or $P_{\text{OSC}}(\boldsymbol{x})$. For $P_{\text{TV}}$, we denote the group of **adjacent elements with the same value** as $z_i^\star$ by $G_i^\star$, $\forall i \in [d]$. Formally, $G_i^\star = [a, b] \cap \mathbb{N}$ with $a \le i \le b$ where $a$ and $b$ are the minimal and maximal indices such that $z_i^\star = z_j^\star$ for all $j \in G_i^\star$. For $P_{\text{OSC}}$, we define $G_i^\star$ as the indices of **elements with the same absolute value** as $z_i^\star$, more formally $G_i^\star = \{j \in [d]: |z_i^\star| = |z_j^\star|\}$. Because $P_{\Delta^d}(\boldsymbol{z}^\star) = \max(\boldsymbol{z}^\star - \theta, 0)$ for some $\theta \in \mathbb{R}$, fusedmax/oscarmax either shift a group's common value or set all its elements to zero.

$\lambda$ controls the trade-off between no fusion (sparsemax) and all elements fused into a single trivial group. While tuning $\lambda$ may improve performance, we observe that $\lambda = 0.1$ (fusedmax) and $\lambda = 0.01$ (oscarmax) are sensible defaults that work well across all tasks and report all our results using them.

**Backward computation.** We already know that the Jacobian of $P_{\Delta^d}$ is the same as that of sparsemax with $\gamma = 1$. Then, by Proposition 2, if we know how to compute the Jacobians of $P_{\text{TV}}$ and $P_{\text{OSC}}$, we can obtain the Jacobians of fusedmax and oscarmax by straightforward application of the chain rule. However, although $P_{\text{TV}}$ and $P_{\text{OSC}}$ can be computed exactly, they lack analytical expressions. We next show that we can nonetheless compute their Jacobians efficiently, without needing to solve a system.

**Proposition 3** *Jacobians of $P_{TV}$ and $P_{OSC}$ (backward computation)*

*Assume $\boldsymbol{z}^\star = P_{TV}(\boldsymbol{x})$ or $P_{OSC}(\boldsymbol{x})$ has been computed. Define the groups derived from $\boldsymbol{z}^\star$ as above.*

*Then,* $[J_{P_{TV}}(\boldsymbol{x})]_{i,j} = \begin{cases} \frac{1}{|G_i^\star|} & \text{if } j \in G_i^\star, \\ 0 & \text{o.w.} \end{cases}$ *and* $[J_{P_{OSC}}(\boldsymbol{x})]_{i,j} = \begin{cases} \frac{\text{sign}(z_i^\star z_j^\star)}{|G_i^\star|} & \text{if } j \in G_i^\star \text{ and } z_i^\star \neq 0, \\ 0 & \text{o.w.} \end{cases}$.

The proof is given in Appendix A.2. Clearly, the structure of these Jacobians permits efficient Jacobian-vector products; we discuss the computational complexity and implementation details in Appendix B. Note that $P_{\text{TV}}$ and $P_{\text{OSC}}$ are differentiable everywhere except at points where groups change. For these points, the same remark as for sparsemax applies, and we can use Clarke's Jacobian.

# 4 Experimental results

We showcase the performance of our attention mechanisms on three challenging natural language tasks: textual entailment, machine translation, and sentence summarization. We rely on available, well-established neural architectures, so as to demonstrate simple drop-in replacement of softmax with structured sparse attention; quite likely, newer task-specific models could lead to further improvement.

## 4.1 Textual entailment (a.k.a. natural language inference) experiments

Textual entailment is the task of deciding, given a text T and an hypothesis H, whether a human reading T is likely to infer that H is true [14]. We use the Stanford Natural Language Inference (SNLI) dataset [8], a collection of 570,000 English sentence pairs. Each pair consists of a sentence and an hypothesis, manually labeled with one of the labels ENTAILMENT, CONTRADICTION, or NEUTRAL.

We use a variant of the neural attention–based classifier proposed for this dataset by [38] and follow the same methodology as [31] in terms of implementation, hyperparameters, and grid search. We employ the CPU implementation provided in [31] and simply replace sparsemax with fusedmax/oscarmax; we observe that training time per epoch is essentially the same for each of the four attention mechanisms (timings and more experimental details in Appendix C.2).

Table 1 shows that, for this task, fusedmax reaches the highest accuracy, and oscarmax slightly outperforms softmax. Furthermore,

Table 1: Textual entailment test accuracy on SNLI [8].

| attention | accuracy |
| --- | --- |
| softmax | 81.66 |
| sparsemax | 82.39 |
| fusedmax | **82.41** |
| oscarmax | 81.76 |

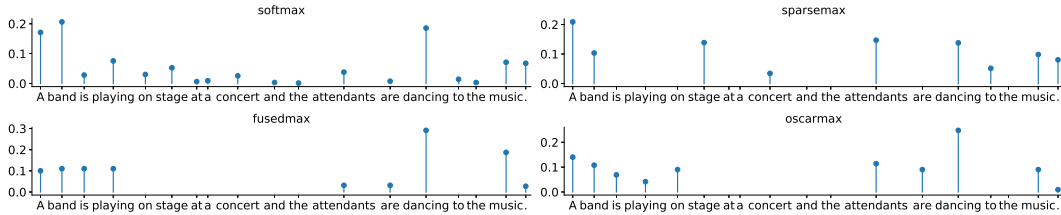

Figure 3: Attention weights when considering the contradicted hypothesis "No one is dancing."

fusedmax results in the most interpretable feature groupings: Figure 3 shows the weights of the neural network's attention to the text, when considering the hypothesis "No one is dancing." In this case, all four models correctly predicted that the text "A band is playing on stage at a concert and the attendants are dancing to the music," denoted along the $x$-axis, **contradicts** the hypothesis, although the attention weights differ. Notably, fusedmax identifies the meaningful segment "band is playing".

## 4.2 Machine translation experiments

Sequence-to-sequence neural machine translation (NMT) has recently become a strong contender in machine translation [2, 29]. In NMT, attention weights can be seen as an *alignment* between source and translated words. To demonstrate the potential of our new attention mechanisms for NMT, we ran experiments on 10 language pairs. We build on OpenNMT-py [24], based on PyTorch [37], with all default hyperparameters (detailed in Appendix C.3), simply replacing softmax with the proposed $\Pi_\Omega$.

OpenNMT-py with softmax attention is optimized for the GPU. Since sparsemax, fusedmax, and oscarmax rely on sorting operations, we implement their computations on the CPU for simplicity, keeping the rest of the pipeline on the GPU. However, we observe that, even with this context switching, the number of tokens processed per second was within ¾ of the softmax pipeline. For sq-pnorm-max, we observe that the projected gradient solver used in the forward pass, unlike the linear system solver used in the backward pass, could become a computational bottleneck. To mitigate this effect, we set the tolerance of the solver's stopping criterion to $10^{-2}$.

Quantitatively, we find that all compared attention mechanisms are always within 1 BLEU score point of the best mechanism (for detailed results, cf. Appendix C.3). This suggests that structured sparsity does not restrict accuracy. However, as illustrated in Figure 4, fusedmax and oscarmax often lead to more interpretable attention alignments, as well as to qualitatively different translations.

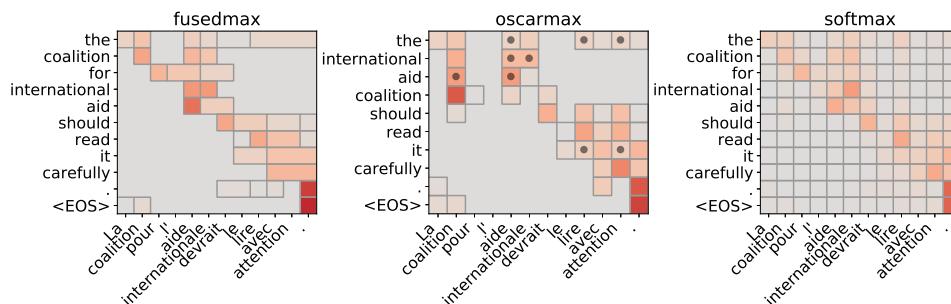

Figure 4: Attention weights for French to English translation, using the conventions of Figure 1. Within a row, weights grouped by oscarmax under the same cluster are denoted by "•". Here, oscarmax finds a slightly more natural English translation. More visulizations are given in Appendix C.3.

## 4.3 Sentence summarization experiments

Attention mechanisms were recently explored for sentence summarization in [39]. To generate sentence-summary pairs at low cost, the authors proposed to use the title of a news article as a noisy summary of the article's leading sentence. They collected 4 million such pairs from the Gigaword dataset and showed that this seemingly simplistic approach leads to models that generalize

Table 2: Sentence summarization results, following the same experimental setting as in [39].

| attention | DUC 2004 | | | Gigaword | | |
|---|---|---|---|---|---|---|
| | ROUGE-1 | ROUGE-2 | ROUGE-L | ROUGE-1 | ROUGE-2 | ROUGE-L |
| softmax | 27.16 | 9.48 | 24.47 | 35.13 | 17.15 | 32.92 |
| sparsemax | 27.69 | 9.55 | 24.96 | 36.04 | **17.78** | 33.64 |
| fusedmax | **28.42** | **9.96** | **25.55** | **36.09** | 17.62 | **33.69** |
| oscarmax | 27.84 | 9.46 | 25.14 | 35.36 | 17.23 | 33.03 |
| sq-pnorm-max | 27.94 | 9.28 | 25.08 | 35.94 | 17.75 | 33.66 |

surprisingly well. We follow their experimental setup and are able to reproduce comparable results to theirs with OpenNMT when using softmax attention. The models we use are the same as in §4.2.

Our evaluation follows [39]: we use the standard DUC 2004 dataset (500 news articles each paired with 4 different human-generated summaries) and a randomly held-out subset of Gigaword, released by [39]. We report results on ROUGE-1, ROUGE-2, and ROUGE-L. Our results, in Table 2, indicate that fusedmax is the best under nearly all metrics, always outperforming softmax. In addition to Figure 1, we exemplify our enhanced interpretability and provide more detailed results in Appendix C.4.

# 5   Related work

**Smoothed max operators.** Replacing the max operator by a differentiable approximation based on the $\log \operatorname{sum} \exp$ has been exploited in numerous works. Regularizing the max operator with a squared 2-norm is less frequent, but has been used to obtain a smoothed multiclass hinge loss [41] or smoothed linear programming relaxations for maximum a-posteriori inference [33]. Our work differs from these in mainly two aspects. First, we are less interested in the max operator itself than in its gradient, which we use as a mapping from $\mathbb{R}^d$ to $\Delta^d$. Second, since we use this mapping in neural networks trained with backpropagation, we study and compute the mapping's Jacobian (the Hessian of a regularized max operator), in contrast with previous works.

**Interpretability, structure and sparsity in neural networks.** Providing interpretations alongside predictions is important for accountability, error analysis and exploratory analysis, among other reasons. Toward this goal, several recent works have been relying on visualizing hidden layer activations [20, 27] and the potential for interpretability provided by attention mechanisms has been noted in multiple works [2, 38, 39]. Our work aims to fulfill this potential by providing a unified framework upon which new interpretable attention mechanisms can be designed, using well-studied tools from the field of structured sparse regularization.

Selecting contiguous text segments for model interpretations is explored in [26], where an *explanation generator* network is proposed for justifying predictions using a fused lasso penalty. However, this network is not an attention mechanism and has its own parameters to be learned. Furthemore, [26] sidesteps the need to backpropagate through the fused lasso, unlike our work, by using a stochastic training approach. In constrast, our attention mechanisms are deterministic and **drop-in replacements** for existing ones. As a consequence, our mechanisms can be coupled with recent research that builds on top of softmax attention, for example in order to incorporate soft prior knowledge about NMT alignment into attention through penalties on the attention weights [12].

A different approach to incorporating structure into attention uses the posterior marginal probabilities from a conditional random field as attention weights [23]. While this approach takes into account structural correlations, the marginal probabilities are generally dense and different from each other. Our proposed mechanisms produce sparse and clustered attention weights, a visible benefit in interpretability. The idea of deriving constrained alternatives to softmax has been independently explored for differentiable easy-first decoding [32]. Finally, sparsity-inducing penalties have been used to obtain convex relaxations of neural networks [5] or to compress models [28, 43, 40]. These works differ from ours, in that sparsity is enforced on the network **parameters**, while our approach can produce sparse and structured **outputs** from neural attention layers.

## 6 Conclusion and future directions

We proposed in this paper a unified regularized framework upon which new attention mechanisms can be designed. To enable such mechanisms to be used in a neural network trained by backpropagation, we demonstrated how to carry out forward and backward computations for general differentiable regularizers. We further developed two new structured attention mechanisms, *fusedmax* and *oscarmax*, and demonstrated that they enhance interpretability while achieving comparable or better accuracy on three diverse and challenging tasks: textual entailment, machine translation, and summarization.

The usefulness of a differentiable mapping from real values to the simplex or to $[0, 1]$ with sparse or structured outputs goes beyond attention mechanisms. We expect that our framework will be useful to sample from categorical distributions using the Gumbel trick [21, 30], as well as for conditional computation [6] or differentiable neural computers [18, 19]. We plan to explore these in future work.

## Acknowledgements

We are grateful to André Martins, Takuma Otsuka, Fabian Pedregosa, Antoine Rolet, Jun Suzuki, and Justine Zhang for helpful discussions. We thank the anonymous reviewers for the valuable feedback.

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
