[Supplementary Material]

# Supplementary material

## A Proofs

### A.1 Proof of Proposition 1

Recall that
$$\Pi_\Omega(\boldsymbol{x}) = \arg\min_{\boldsymbol{y} \in \Delta^d} f(\boldsymbol{y}), \quad \text{where} \quad f(\boldsymbol{y}) := \gamma\Omega(\boldsymbol{y}) - \boldsymbol{y}^\mathrm{T}\boldsymbol{x}.$$

At an optimal solution, we have the fixed point iteration [36, §4.2]
$$\boldsymbol{y}^* = P_{\Delta^d}(\boldsymbol{y}^\star - \nabla f(\boldsymbol{y}^\star)). \tag{2}$$

Seeing $\boldsymbol{y}^\star$ as a function of $\boldsymbol{x}$, and $P_{\Delta^d}$ and $\nabla f$ as functions of their inputs, we can apply the chain rule to (2) to obtain
$$J_{\Pi_\Omega}(\boldsymbol{x}) = J_{P_{\Delta^d}}\left(\boldsymbol{y}^\star - \nabla f(\boldsymbol{y}^\star)\right)\left(J_{\Pi_\Omega}(\boldsymbol{x}) - J_{\nabla f \circ \boldsymbol{y}^\star}(\boldsymbol{x})\right). \tag{3}$$

Applying the chain rule once again to $\nabla f(\boldsymbol{y}^\star) = \gamma\nabla\Omega(\boldsymbol{y}^\star) - \boldsymbol{x}$, we obtain
$$\begin{aligned} J_{\nabla f \circ \boldsymbol{y}^\star}(\boldsymbol{x}) &= \gamma J_{\nabla\Omega}(\boldsymbol{y}^\star)J_{\Pi_\Omega}(\boldsymbol{x}) - I \\ &= \gamma H_\Omega(\boldsymbol{y}^\star)J_{\Pi_\Omega}(\boldsymbol{x}) - I. \end{aligned}$$

Plugging this into (3) and re-arranging, we obtain
$$\left(I + A(B - I)\right)J_{\Pi_\Omega} = A,$$

where we defined the shorthands
$$J_{\Pi_\Omega} := J_{\Pi_\Omega}(\boldsymbol{x}), \quad A := J_{P_{\Delta^d}}(\boldsymbol{y}^\star - \gamma\nabla\Omega(\boldsymbol{y}^\star) + \boldsymbol{x}) \quad \text{and} \quad B := \gamma H_\Omega(\boldsymbol{y}^\star).$$

### A.2 Proof of Proposition 3

**Proof outline.** Let $\boldsymbol{z}^\star = P_{\mathrm{TV}}(\boldsymbol{x})$ or $P_{\mathrm{OSC}}(\boldsymbol{x})$. We use the optimality conditions of $P_{\mathrm{TV}}$, respectively $P_{\mathrm{OSC}}$ in order to express $\boldsymbol{z}^\star$ as an explicit function of $\boldsymbol{x}$. Then, obtaining the Jacobians of $P_{\mathrm{TV}}(\boldsymbol{x})$ and $P_{\mathrm{OSC}}(\boldsymbol{x})$ follows by application of the chain rule to the two expressions. We discuss the proof for points where $P_{\mathrm{TV}}$ and $P_{\mathrm{OSC}}$ are differentiable; on the (zero-measure) set of nondifferentiable points (i.e. where the group structure changes) we may take one of Clarke's generalized gradients [11].

**Jacobian of $P_{\mathrm{TV}}$.**

**Lemma 1** *Let $\boldsymbol{z}^\star = P_{\mathrm{TV}}(\boldsymbol{x}) \in \mathbb{R}^d$ and $G_i^\star$ be the set of indices around $i$ with the same value at the optimum, as defined in §3.2. Then, we have*
$$z_i^\star = \frac{\sum_{j \in G_i^\star} x_j + \lambda(s_{a_i} - s_{b_i})}{|G_i^\star|}, \tag{4}$$

*where $a_i = \min G_i^\star, b_i = \max G_i^\star$ are the boundaries of segment $G_i^\star$, and*
$$s_{a_i} = \begin{cases} 0 & \text{if } a = 1, \\ \mathrm{sign}(z_{a_i-1}^\star - z_i^\star) & \text{if } a > 1 \end{cases} \quad \text{and} \quad s_{b_i} = \begin{cases} 0 & \text{if } b = d, \\ \mathrm{sign}(z_i^\star - z_{b_i+1}^\star) & \text{if } b < d \end{cases}.$$

To prove Lemma 1, we make use of the optimality conditions of the fused lasso proximity operator [17, Equation 27], which state that $\boldsymbol{z}^\star$ satisfies
$$z_j^\star - x_j + \lambda(t_j - t_{j+1}) = 0, \quad \text{where} \quad t_j \in \begin{cases} \{0\} & \text{if } i \in \{1, d\}, \\ \{\mathrm{sign}(z_j^\star - z_{j-1}^\star)\} & \text{if } z_j^\star \neq z_{j-1}^\star, \quad \forall j \in [d]. \\ [-1, 1] & \text{o.w.} \end{cases} \tag{5}$$

The optimality conditions (5) form a system with unknowns $z_j^\star, t_j$ for $j \in [d]$. To express $\boldsymbol{z}^\star$ as a function of $\boldsymbol{x}$, we shall now proceed to eliminate the unknowns $t_j$.

Let us focus on a particular segment $G_i^\star$. For readability, we drop the segment index $i$ and use the shorthands $z := z_i^\star, a := a_i$, and $b := b_i$. By definition, $a$ and $b$ satisfy

$$z_j^\star = z \quad \forall a \le j \le b, \qquad z_{a-1}^\star \ne z \text{ if } a > 1, \qquad z_{b+1}^\star \ne z \text{ if } b < d.$$

It immediately follows from the definition of $t_j$ in (5) that

$$t_a = \begin{cases} 0 & \text{if } a = 1, \\ \text{sign}(z - z_{a-1}^\star) & \text{if } a > 1 \end{cases} \quad \text{and} \quad t_{b+1} = \begin{cases} 0 & \text{if } b = d, \\ \text{sign}(z_{b+1}^\star - z) & \text{if } b < d \end{cases}.$$

In other words, the unknowns $t_a$ and $t_b$ are already uniquely determined. To emphasize that they are known, we introduce $s_a := t_a$ and $s_b := t_{b+1}$, leaving $t_j$ only unknown for $a < j \le b$.

By rearranging the optimality conditions (5) we obtain the recursion

$$\lambda t_j = x_j - z + \lambda t_{j+1} \quad \forall a \le j \le b.$$

We start with the first equation in the segment (at $j = a$), and unroll the recursion until reaching the stopping condition $j = b$.

$$\begin{aligned} \lambda s_a &= x_a - z + \lambda t_{a+1} \\ &= x_a - z + x_{a+1} - z + \cdots + x_b - z + \lambda s_b \\ &= \sum_{k=a}^b x_k - (b - a + 1)z + \lambda s_b \end{aligned}$$

Rearranging the terms, we obtain the expression

$$z = \frac{\sum_{k=a}^b x_k + \lambda(s_b - s_a)}{b - a + 1}.$$

Applying this calculation to each segment in $\boldsymbol{z}^\star$ yields the desired result. $\qquad\square$

The proof of Proposition 3 follows by applying the chain rule to (4), noting that the groups $G_i^*$ are constant within a neighborhood of $\boldsymbol{x}$ (observation also used for OSCAR in [7]). Therefore, for $P_{\text{TV}}$,

$$\frac{\partial z_i^\star}{\partial x_j} = \frac{1}{|G_i^\star|} \left( \sum_{k \in G_i^\star} \frac{\partial x_k}{\partial x_j} + \lambda \left( \frac{\partial s_b}{\partial x_j} - \frac{\partial s_a}{\partial x_j} \right) \right).$$

Since $s_b$ and $s_a$ are either constant or sign functions w.r.t. $\boldsymbol{x}$, their partial derivatives are 0, and thus

$$\frac{\partial z_i^\star}{\partial x_j} = \begin{cases} \frac{1}{|G_i^\star|} & \text{if } j \in G_i^\star, \\ 0 & \text{o.w.} \end{cases}.$$

**Jacobian of $P_{\text{OSC}}$.**

**Lemma 2** *([47, Theorem 1], [49, Proposition 3]) Let $\boldsymbol{z}^\star = P_{OSC}(\boldsymbol{x}) \in \mathbb{R}^d$ and $G_i^\star$ be the set of indices around $i$ with the same value at the optimum: $G_i^\star = \{j \in [d] : |z_i^\star| = |z_j^\star|\}$. Then, we have*

$$z_i^\star = \text{sign}(x_i) \max \left( \frac{\sum_{j \in G_i^\star} |x_j|}{|G_i^\star|} - w_i, 0 \right), \tag{6}$$

*where $w_i = \lambda \left( d - \frac{u_i + v_i}{2} \right)$,    $u_i = \left| \{j \in [d] : |z_j^\star| < |z_i^\star|\} \right|$,    $v_i = u_i + |G_i^\star|$.*

Lemma 2 is a simple reformulation of Theorem 1, part *ii* from [47]. With the same observation that the induced groups do not change within a neighborhood of $\boldsymbol{x}$, we may differentiate (6) to obtain

$$
\frac{\partial z_i^\star}{\partial x_j} = \begin{cases} 0 & \text{if } z_i^\star = 0, \\ \dfrac{\operatorname{sign}(x_i)}{|G_i^\star|} \displaystyle\sum_{k \in G_i^\star} \frac{\partial |x_k|}{\partial x_k} \frac{\partial x_k}{\partial x_j} - \frac{\partial w_i}{\partial x_j} & \text{o.w.} \end{cases} .
$$

Noting that $\frac{\partial w_i}{\partial x_j} = 0$, as $w_i$ is derived only from group indices and the term $\frac{\partial |x_k|}{\partial x_k} \frac{\partial x_k}{\partial x_j}$ either vanishes (when $k \neq j$) or else equals $\operatorname{sign}(x_j)$ with $x_j \neq 0$, we substitute $\operatorname{sign}(z_j^\star)$ for $\operatorname{sign}(x_j)$ [47] to get

$$
\frac{\partial z_i^\star}{\partial x_j} = \begin{cases} \dfrac{\operatorname{sign}(z_i^\star z_j^\star)}{|G_i^\star|} & \text{if } j \in G_i^\star \text{ and } z_i^\star \neq 0, \\ 0 & \text{o.w.} \end{cases} .
$$

# B  Computational complexity and implementation details

## B.1  Sparsemax

Computing the forward and backward pass of sparsemax is a compositional building block in fusedmax, oscarmax, as well as in the general case; for this reason, we discuss it before the others.

**Forward pass.** The problem is exactly the Euclidean projection on the simplex, which can be computed exactly in worst-case $\mathcal{O}(d \log d)$ due to the required sort [31, 34, 15], or in expected $\mathcal{O}(d)$ time using a pivot algorithm similar to median finding [15]. Our implementation is based on sorting.

**Backward pass.** From [31] we have that the result of a Jacobian-vector product $J_{\Pi_\Omega} v$ has the same sparsity pattern as $\boldsymbol{y}^\star$. If we denote the number of nonzero elements of $\boldsymbol{x}$ by $\mathrm{nnz}(\boldsymbol{x})$, we can see that $\hat{v}$ in [31, eq. 14], and thus the Jacobian-vector product itself, can be computed in $\mathcal{O}(\mathrm{nnz}(\boldsymbol{y}^\star))$.

## B.2  Fusedmax

We implement fusedmax as the composition of the fused lasso proximal operator with sparsemax.

**Forward pass.** We need to solve the proximal operator of fused lasso. The algorithm we use is $\mathcal{O}(d^2)$ in the worst case, but has strong performance on realistic benchmarks, close to $\mathcal{O}(d)$ [13].

**Backward pass.** Due to the structure of the Jacobian and the locality of fused groups, Jacobian-vector products $J_{\Pi_\Omega} v$ can be computed in $\mathcal{O}(d)$ using a simple algorithm that iterates over the output $\boldsymbol{y}^\star$ and the vector $\boldsymbol{v}$ simultaneously, averaging the elements of $\boldsymbol{v}$ whose indices map to fused elements of $\boldsymbol{y}^\star$. Since only consecutive elements can be fused, this amounts to resetting to a new group as soon as we encounter an index $i$ such that $y_i^\star \neq y_{i-1}^\star$.

## B.3  Oscarmax

We implement oscarmax as the composition of the OSCAR proximal operator with sparsemax.

**Forward pass.** The proximal operator of the OSCAR penalty can be computed in $\mathcal{O}(d \log d)$ as a particular case of the ordered weighted $\ell_1$ (OWL) proximal operator, using an algorithm involving a sort followed by isotonic regression [48].

**Backward pass.** The algorithm is similar in spirit to fusedmax, but because groups can reach across non-adjacent indices, a single pass is not sufficient. With no other information other than $\boldsymbol{y}^\star$, the backward pass can be computed in $\mathcal{O}(d \log d)$ using a stable sort followed by a linear-time pass for finding groups. Further optimization is possible if group indices may be saved from the forward pass.

## B.4  General case and sq-pnorm-max

**Forward pass.** For general $\Pi_\Omega$ we may use any projected gradient solver; we choose FISTA [4]. Each iteration requires a projection onto the simplex; in the case of sq-pnorm-max, this dominates every iteration, leading to a complexity of $\mathcal{O}(td \log d)$ where $t$ is the number of iterations performed.

**Backward pass.** To compute Jacobian-vector products we solve the linear system from Proposition 1: $(I + A(B - I))(J_{\Pi_\Omega}\boldsymbol{v}) = A\boldsymbol{v}$. This is a $d \times d$ system, which at first sight suggests a complexity of $\mathcal{O}(d^3)$. However, we can use the structure of $A$ to solve it more efficiently.

The matrix $A$ is defined as $A := J_{P_{\Delta d}}(\boldsymbol{y}^\star - \nabla f(\boldsymbol{y}^\star))$. As a sparsemax Jacobian, $A$ is row- and column-sparse, and uniquely defined by its sparsity pattern. By splitting the system into equations corresponding to zero and nonzero rows of $A$, we obtain that the solution $J_{\Pi_\Omega}\boldsymbol{v}$ must have the same sparsity pattern as the row-sparsity of $A$, therefore we only need to solve a subset of the system. From the fixed-point iteration $\boldsymbol{y}^* = P_{\Delta d}(\boldsymbol{y}^\star - \nabla f(\boldsymbol{y}^\star))$, we have that the row-sparsity of $A$ is the same as the sparsity of the forward pass solution $\boldsymbol{y}^*$. The backward pass complexity is thus $\mathcal{O}(\mathrm{nnz}(\boldsymbol{y}^*)^3)$.

## C    Additional experimental results

### C.1    Visualizing attention mappings in 3-d

Figure 5: 3-d visualization of $\Pi_\Omega([t_1, t_2, 0])_2$ for several proposed and existing mappings $\Pi_\Omega$. sq-pnorm-max with $p = 1.5$ resembles sparsemax but with smoother transitions. The proposed structured attention mechanisms, fusedmax and oscarmax, exhibit plateaus and ridges in areas where weights become fused together. We set $\gamma = 1$ and $\lambda = 0.2$.

### C.2    Textual entailment results

**Experimental setup.** We build upon the implementation from [31], which is a slight variation of the attention model from [38], using GRUs instead of LSTMs. The GRUs encoding the premise and hypothesis have separate parameters, but the hypothesis GRU is initialized with the last state of the premise GRU. We use the same settings and methodology as [31]: we use fixed 300-dimensional GloVe vectors, we train for 200 epochs using ADAM with learning rate $3 \cdot 10^{-4}$, we use a drop-out probability of 0.1, and we choose an $l_2$ regularization coefficient from $\{0, 10^{-4}, 3 \cdot 10^{-4}, 10^{-3}\}$. Experiments are performed on machines with 2×Xeon X5675 3.06GHz CPUs and 96GB RAM.

**Dataset and preprocessing.** We use the SNLI v1 dataset [8]. We apply the minimal preprocessing from [31], skipping sentence pairs with missing labels and using the provided tokenization. This results in a training set of 549,367 sentence pairs, a development set of 9,842 sentence pairs and a test set of 9,824 sentence pairs. We report timing measurements in Table 3 and visualizations of the produced attention weights in Figure 6.

### C.3    Machine translation results

**Experimental setup.** Because our goal is to demonstrate that our attention mechanisms can be drop-in replacements for existing ones, we focus on OpenNMT-py with default settings for all of our

Figure 6: Attention weights on several examples also used in [38, 31]. The hypotheses considered are "Two mimes sit in complete silence." (top), "A boy is riding an animal." (left), and "Two dogs swim in the lake." (right). All attention mechanisms result in correct classifications (top: contradiction; left: entailment; right: contradiction). As can be seen, fusedmax prefers contiguous support segments even when not all weights are tied.

sequence-to-sequence experiments. These defaults are: an unidirectional LSTM, 500 dimensions for the word vectors and for the LSTM hidden representations, drop-out probability of 0.3, global attention, and input-feeding [29]. Following the default, we train our models for 13 epochs with stochastic gradient updates (batches of size 64 and initial learning rate of 1, halved every epoch after the 8[th]). Weights (including word embeddings) are initialized uniformly over $[-0.1, 0.1]$, and gradients are normalized to have norm 5 if their norm exceeds this value. For test scores and visualizations, we use the model snapshot at the epoch with the highest validation set accuracy. All of the experiments in this section are performed on machines equiped with Xeon E5 CPUs and Nvidia Tesla K80 GPUs.

**Datasets.** We employ training and test datasets from multiple sources.

| attention | time per epoch |
|---|---|
| softmax | 1h 26m 40s ± 51s |
| sparsemax | 1h 24m 21s ± 54s |
| fusedmax | 1h 23m 58s ± 50s |
| oscarmax | 1h 23m 19s ± 50s |

Table 3: Timing results for training textual entailment on SNLI, using the implementation and experimental setup from [31]. With this C++ CPU implementation, fusedmax and oscarmax are as fast as sparsemax, and all three sparse attention mechanisms are slightly faster than softmax.

- BENCHMARK: Training, validation, and test data from the NMT-Benchmark project (`http://scorer.nmt-benchmark.net/`). All languages have ~1M training sentence pairs, and equal validation and test sets of size 1K (French) and 2K (Italian, Dutch and Swedish).

- BENCHMARK$^+$: Training and validation data as above, but testing on all available *newstest* data. For Italian we use the 2009 data (~2.5K sentence pairs), and for French we concatenate 2009–2014 (~11K sentence pairs).

- WMT16, WMT17: Translation tasks at the first and second ACL Conferences for Machine Translation, available at `http://www.statmt.org/wmt16/translation-task.html` and `http://www.statmt.org/wmt17/translation-task.html`. Training, validation, and test sizes are, approximately, for Romanian 400K/2K/2K, for German 5.8M/6K/3K, for Finnish 2.6M/2K/2K, for Latvian 4.5M/2K/2K, and for Turkish 207K/1K/3K.

We use the preprocessing scripts from Moses [25] for tokenization, and, where needed, SGML parsing. We limit source and target vocabulary sizes to 50K lower-cased tokens and prune sentences longer than 50 tokens at training time and 100 tokens at test time. We do not perform recasing.

We report BLEU scores in Table 4 and showcase the enhanced interpretability induced by our proposed attention mechanisms in Figure 7. Timing measurements can be found in Table 5.

Table 4: Neural machine translation results: tokenized BLEU scores on test data.

| | BENCHMARK | | | | BENCHMARK$^+$ | | WMT16 | WMT17 | | | |
|---|---|---|---|---|---|---|---|---|---|---|---|
| | fr | it | nl | sv | fr | it | ro | de | fi | lv | tr |
| **from English** | | | | | | | | | | | |
| softmax | 36.94 | 37.20 | 36.12 | 34.97 | 27.13 | **24.86** | 17.71 | 22.32 | 14.54 | 11.02 | **11.95** |
| sparsemax | 37.03 | 37.21 | 36.12 | **35.09** | 26.99 | 24.49 | 17.61 | **22.43** | **14.85** | 11.07 | 11.66 |
| fusedmax | 37.08 | 36.73 | 36.04 | 34.30 | 26.89 | 24.47 | 17.19 | 22.25 | 14.28 | **11.27** | 11.32 |
| oscarmax | 36.66 | 36.89 | 35.96 | 34.86 | 27.02 | 24.76 | 17.26 | 22.42 | 14.02 | 11.19 | 11.63 |
| sq-pnorm-max | **37.16** | **37.39** | **36.21** | 34.63 | **27.25** | 24.56 | **17.80** | —— | 14.45 | —— | 11.58 |
| **to English** | | | | | | | | | | | |
| softmax | 36.79 | 39.95 | 40.06 | 37.96 | 25.72 | 25.37 | 17.86 | **25.82** | 15.11 | 13.60 | 11.78 |
| sparsemax | **36.91** | 40.13 | 40.25 | 38.09 | 25.97 | 25.62 | 17.46 | 25.76 | 14.95 | 13.59 | **12.04** |
| fusedmax | 36.64 | 39.64 | 39.87 | 37.83 | 25.72 | 25.41 | **18.29** | 25.58 | 15.08 | 13.53 | 11.91 |
| oscarmax | 36.90 | 40.05 | 40.17 | **38.12** | **26.13** | 25.65 | 17.89 | 25.69 | 14.94 | **13.71** | 11.70 |
| sq-pnorm-max | 36.84 | **40.23** | **40.48** | **38.12** | 25.72 | **25.70** | 17.44 | —— | **15.20** | —— | 11.93 |

| attention | time per epoch |
|---|---|
| softmax | 2h |
| sparsemax | 2h 18m |
| fusedmax | 3h 5m |
| oscarmax | 3h 25m |
| sq-pnorm-max | 7h 5m |

Table 5: Timing results for French-to-English translation using OpenNMT-py (all standard errors are under 2 minutes). For simplicity, all attention mechanisms, except softmax, are implemented on the CPU, thus incurring memory copies in both directions. (The rest of the pipeline runs on the GPU.) Even without special optimization, sparsemax, fusedmax, and oscarmax are practical, taking within 1.75x the training time of a softmax model on the GPU.

Figure 7: Attention alignment examples for French-to-English translation, following the conventions of Figure 1. "@-@" denotes a hyphen not separated by spaces. When oscarmax induces multiple clusters, we denote them using different bullets (e.g., ●, ▲, ■). Fusedmax often selects meaningful grammatical segments, such as "est consacré," as well as determiner-noun constructions.

Figure 7 (continued): Further translation examples from French to English.

## C.4   Sentence summarization results

**Experimental setup and data.** We use the exact same experimental setup and preprocessing as for machine translation, described in Appendix C.3. We use the preprocessed Gigaword sentence summarization dataset, made available by the authors of [39] at `https://github.com/harvardnlp/sent-summary`. Since, unlike [39], we do not perform any tuning on DUC-2003, we can report results on this dataset, as well. We observe that the simple sequence-to-sequence model is able to keep summaries short without any explicit constraints, informed only through training data statistics; therefore, in this section, we also report results without output truncation at 75 bytes (Table 6). We also provide precision and recall scores for ROUGE-L in Table 7. Finally, we provide attention weights plots for all studied attention mechanisms and a number of validation set examples in Figure 8.

Table 6: Sentence summarization $F_1$ scores for several ROUGE variations.

| attention | Truncated | | | | Not truncated | | | |
|---|---|---|---|---|---|---|---|---|
| | ROUGE-1 | ROUGE-2 | ROUGE-L | ROUGE-W$_{1.2}$ | ROUGE-1 | ROUGE-2 | ROUGE-L | ROUGE-W$_{1.2}$ |
| **DUC 2003** | | | | | | | | |
| softmax | 26.63 | 8.72 | 23.87 | 16.95 | 27.06 | 8.86 | 24.23 | 17.02 |
| sparsemax | 26.54 | 8.78 | 23.89 | 16.93 | 26.95 | 8.94 | 24.21 | 16.99 |
| fusedmax | **27.12** | 8.93 | **24.39** | **17.28** | 27.48 | 9.04 | **24.66** | **17.30** |
| oscarmax | 26.72 | **9.08** | 24.02 | 17.06 | 27.11 | **9.23** | 24.32 | 17.10 |
| sq-pnorm-max | 26.55 | 8.77 | 23.78 | 16.87 | 26.92 | 8.89 | 24.07 | 16.92 |
| **DUC 2004** | | | | | | | | |
| softmax | 27.16 | 9.48 | 24.47 | 17.14 | 27.25 | 9.52 | 24.55 | 17.20 |
| sparsemax | 27.69 | 9.55 | 24.96 | 17.44 | 27.77 | 9.61 | 25.02 | 17.48 |
| fusedmax | **28.42** | **9.96** | **25.55** | **17.78** | **28.43** | **9.96** | **25.55** | **17.79** |
| oscarmax | 27.84 | 9.46 | 25.14 | 17.55 | 27.88 | 9.47 | 25.17 | 17.57 |
| sq-pnorm-max | 27.94 | 9.28 | 25.08 | 17.49 | 28.01 | 9.30 | 25.13 | 17.52 |
| **Gigaword** | | | | | | | | |
| softmax | 35.13 | 17.15 | 32.92 | 24.17 | 35.01 | 17.10 | 32.77 | 24.00 |
| sparsemax | 36.04 | **17.78** | 33.64 | 24.69 | 35.97 | **17.75** | 33.54 | **24.55** |
| fusedmax | **36.09** | 17.62 | **33.69** | 24.69 | **35.98** | 17.60 | **33.59** | 24.54 |
| oscarmax | 35.36 | 17.23 | 33.03 | 24.25 | 35.26 | 17.20 | 32.92 | 24.10 |
| sq-pnorm-max | 35.94 | 17.75 | 33.66 | **24.71** | 35.86 | 17.73 | 33.54 | **24.55** |

Table 7: Sentence summarization: ROUGE-L precision, recall and F-scores.

| attention | Truncated | | | Not truncated | | |
|---|---|---|---|---|---|---|
| | $P$ | $R$ | $F_1$ | $P$ | $R$ | $F_1$ |
| **DUC 2003** | | | | | | |
| softmax | 29.57 | 20.67 | 23.87 | 30.40 | 20.80 | 24.23 |
| sparsemax | 29.59 | 20.58 | 23.89 | 30.37 | 20.68 | 24.21 |
| fusedmax | **30.02** | **21.11** | **24.39** | **30.75** | **21.15** | **24.66** |
| oscarmax | 29.64 | 20.78 | 24.02 | 30.40 | 20.87 | 24.32 |
| sq-pnorm-max | 29.45 | 20.50 | 23.78 | 30.23 | 20.56 | 24.07 |
| **DUC 2004** | | | | | | |
| softmax | 30.54 | 21.00 | 24.47 | 30.59 | 21.13 | 24.55 |
| sparsemax | 30.99 | 21.57 | 24.96 | 31.03 | 21.64 | 25.02 |
| fusedmax | **32.19** | **21.80** | **25.55** | **32.19** | **21.81** | **25.55** |
| oscarmax | 31.89 | 21.46 | 25.14 | 31.91 | 21.51 | 25.17 |
| sq-pnorm-max | 31.42 | 21.55 | 25.08 | 31.46 | 21.63 | 25.13 |
| **Gigaword** | | | | | | |
| softmax | 36.43 | 31.67 | 32.92 | 36.61 | 31.54 | 32.77 |
| sparsemax | 37.32 | 32.18 | 33.64 | 37.54 | 32.07 | 33.54 |
| fusedmax | **37.44** | 32.15 | **33.69** | **37.68** | 32.01 | **33.59** |
| oscarmax | 36.40 | 31.78 | 33.03 | 36.61 | 31.67 | 32.92 |
| sq-pnorm-max | 37.12 | **32.37** | 33.66 | 37.31 | **32.26** | 33.54 |

Figure 8: Summarization attention examples. The 1-d TV prior of fusedmax captures well the intuition of aligning long input spans with single expressive words, as supported by ROUGE scores.

Figure 8 (continued): Summarization attention examples. Here, fusedmax recovers a longer but arguably better summary, identifying a separate but important part of the input sentence.

Figure 8 (continued): Summarization attention examples. Here, fusedmax and oscarmax produce a considerably shorter summary.