[Reviews · NeurIPS 2017]

Reviewer 1



The authors investigate different mechanisms for attention in neural networks. Classical techniques are based on the softmax function where all elements in the input always make at least a small contribution to the decision. To this end, sparse-max [29] was recently shown to work well. It focuses on a subset of the domain, treating the remaining part as `not contributing’. Here, the authors generalize this concept by applying a general smoothing technique which is based on conjugate functions of the max operator. The authors then show how to compute the required derivatives for training and how to employ structured regularization variants. The approach is validated on textual entailment (Stanford Natural Language Inference dataset), machine translation and sentence summarization (DUC 2004 and Gigaword dataset). The proposed regularization terms are explored on those tasks and advantages are demonstrated compared to classical softmax regularization. Strength: The paper investigates interesting techniques and convincingly evaluates them on a variety of tasks. Weaknesses: The authors argue that the proposed techniques `often lead to more interpretable attention alignments’. I wonder whether this is quantifiable? Using computer vision based models, attention mechanisms don’t necessarily correlate well with human attention. Can the authors comment on those effects for NLP? Although mentioned in the paper, the authors don’t discuss the computational implications of the more complex regularization mechanisms explicitly. I think a dedicated section could provide additional insights for a reader.

Reviewer 2



This paper presents a number of sparsifying alternatives to the softmax operator that can be straightforwardly integrated in the backpropagation of neural networks. These techniques can be used in general to approximate categorical inference in neural networks, and in particular to implement attention. The paper extends work such as sparsemax from its reference [29]. I believe this is a strong paper with significant contributions and thorough justifications. The paper presents: 1. The gradient of a general regularization of the max operator (Section 2.2). 2. Softmax and sparsemax as examples (2.3). 3. An original derivation of the Jacobian of any differentiable regularizer, and an example with the squared p-norm (3.1). 4. Two examples with structured regularizers, with algorithms for their computation and an original derivation of their Jacobians (3.2). One of them (fusedmax) is a novel use of the fused Lasso [39] as an attention mechanism. One may find a limitation of this paper in the fact that the experiments are carried out using networks that are not anymore the state of the art. As such, the paper does not claim any improvements over state-of-the-art accuracies. However, in my opinion this is not important since the aim is to replace the softmax operator in a variety of plausible architectures. The related work section is very clear and current, including a reference ([21]) on structured attention from ICLR 2017 that had immediately come to my mind. The structure discussed in this reference is more general; however, the contributions are very different. The paper also contains rich and descriptive supplemental material.

Reviewer 3



Summary ======= This paper presents a framework for implementing different sparse attention mechanisms by regularizing the max operator using convex functions. As a result, softmax and sparsemax are derived as special cases of this framework. Furthermore, two new sparse attention mechanisms are introduced that allow the model to learn to pay the same attention to contiguous spans. My concerns are regarding to the motivation of interpretability, as well as the baseline attention models. However, the paper is very well presented and the framework is a notable contribution that I believe will be useful for researchers working with attention mechanisms. Strengths ========= - The provided framework encompasses softmax and sparsemax as special cases. - It is a drop-in replacement for softmax, so it can be readily used for extending a variety of existing neural networks that employ a softmax attention mechanism. - There seems to only be a small computational overhead (~25% increase in runtime). - While the attention weights produced by the sparse attention models are slightly more interpretable, using such sparse attention didn't hurt performance on RTE, MT and sentence summarization tasks. Weaknesses ========== - Looking at Fig 1 and playing Devil's advocate, I would argue that interpretability doesn't really seem to improve that much with the variants of sparse attention presented here -- I could just threshold the weights of a normal softmax attention mechanism (or increase the contrast in Fig 1) and I would get a similar outcome. I do see however that with span-based sparse attention things start to look more interpretable but I have the impression that the three tasks are not the best fit for it. For instance, Q&A on SQuAD might have been a more interesting task as it asks for a span in a paragraph as the answer for a given question. This might have alleviated another weakness of the evaluation, namely that we don't see strong improvements when using span-based sparse attention. So right now the main takeaway is that we can obtain similar performance with slightly improved interpretability -- where I am still unsure how practically relevant the latter is. - Related to the point above, I do believe it is important to point out that interpretability of attention weights is problematic as the attentive RTE and MT (and maybe the summarization model too?) are not solely relying on the context vector that is obtained from the attention mechanism. Hence, any conclusions drawn from looking at the attention weights should be taken with a big grain of salt. - Moreover, the RTE attention baseline seems weak (though I understand that model is inherited from sparsemax). [35] report a test set accuracy of 83.5% whereas the baseline attentive RTE model here only achieves 81.66%, and the best performance for fusedmax reported here is 82.41%. In fact, what I would have liked to see is a comparison of a state-of-the-art attentive RTE/MT model with a drop-in replacement of fusedmax/oscarmax and an improvement in accuracy/BLEU. Minor Comments ============== - I think it would be great if the limitations of this framework could be highlighted a bit more. For instance, I am wondering whether your framework could also support more global structured penalties such as the prior in MT that most of the attention should be along the diagonal and my hunch is that this would be difficult. See Cohn, Trevor, et al. "Incorporating structural alignment biases into an attentional neural translation model." arXiv preprint arXiv:1601.01085 (2016). - I believe in section 2 it could be made clearer when gradients are calculated by a solver and when not.